# Association between Bone Turnover Markers and Fracture Healing in Long Bone Non-Union: A Systematic Review

**DOI:** 10.3390/jcm13082333

**Published:** 2024-04-17

**Authors:** Francesca Perut, Laura Roncuzzi, Enrique Gómez-Barrena, Nicola Baldini

**Affiliations:** 1Biomedical Science and Technologies and Nanobiotechnology Laboratory, IRCCS Istituto Ortopedico Rizzoli, 40136 Bologna, Italy; laura.roncuzzi@ior.it (L.R.); nicola.baldini@ior.it (N.B.); 2Department of Orthopedic Surgery and Traumatology, Hospital Universitario La Paz-IdiPAZ, 28046 Madrid, Spain; egomezb@salud.madrid.org; 3Facultad de Medicina, Universidad Autónoma de Madrid, 28029 Madrid, Spain; 4Department of Biomedical and Neuromotor Sciences, University of Bologna, 40136 Bologna, Italy

**Keywords:** fracture healing, non-union, bone turnover markers, delayed union, prognostic, biomarkers

## Abstract

**Background**: Fracture healing is a very complex and well-orchestrated regenerative process involving many cell types and molecular pathways. Despite the high efficiency of this process, unsatisfying healing outcomes, such as non-union, occur for approximately 5–10% of long bone fractures. Although there is an obvious need to identify markers to monitor the healing process and to predict a potential failure in callus formation to heal the fracture, circulating bone turnover markers’ (BTMs) utility as biomarkers in association with radiographic and clinical examination still lacks evidence so far. **Methods**: A systematic review on the association between BTMs changes and fracture healing in long bone non-union was performed following PRISMA guidelines. The research papers were identified via the PubMed, Cochrane, Cinahl, Web of Science, Scopus, and Embase databases. Studies in which the failure of fracture healing was associated with osteoporosis or genetic disorders were not included. **Results**: A total of 172 studies were collected and, given the inclusion criteria, 14 manuscripts were included in this review. Changes in circulating BTMs levels were detected during the healing process and across groups (healed vs. non-union patients and healthy vs. patients with non-union). However, we found high heterogeneity in patients’ characteristics (fracture site, gender, and age) and in sample scheduling, which made it impossible to perform a meta-analysis. **Conclusions**: Clinical findings and radiographic features remain the two important components of non-union diagnosis so far. We suggest improving blood sample standardization and clinical data collection in future research to lay the foundations for the effective use of BTMs as tools for diagnosing non-union.

## 1. Introduction

Long bone fractures, including those of the femur, tibia and humerus, represent some of the most frequent trauma-derived consequences worldwide [1]. Non-union fractures are well-defined medical conditions that represent the inability of bone to heal without further intervention, which is usually identified based on radiological, clinical, and temporal criteria. The patients complain from pain at the fracture site, inability to bear weight, associated disability, and prolonged hospitalization, resulting in increased health expenses [2]. The Food and Drug Administration (USA) recommends defining the non-union in clinical trials evaluating bone substitutes as the absence of bone bridging and the presence of a fracture line nine months after the original fracture, with a lack of progressive signs of bone healing over three consecutive months [3,4]. Delayed union generally refers to a fracture that has not healed after three to six months, while fractures that fail to heal after the first non-union therapeutic procedure are defined as recalcitrant non-unions [5].

Fracture healing generally depends on primary adequate mechanical stabilization of the fracture site, which avoids the displacement of bone ends, thus facilitating efficient bone regeneration from the hematoma towards spontaneous healing in 90–95% of cases. However, some conditions ascribable to patients (i.e., age, gender, concomitant diseases (i.e., diabetes), type of fracture, current smoking, obesity, alcohol abuse) and/or biological factors (i.e., infections, peripheral vascular disease, etc.) may affect the healing process [6,7]. Moreover, the use of certain drugs, such as bisphosphonates or chronic administration of non-steroid anti-inflammatory drugs, may delay fracture healing [8,9].

Non-union diagnosis is currently based on clinical and radiographic evaluation. Indeed, pain and loss of function with or without mobility at the fracture site are clinical signs that suggest the presence of pseudarthrosis. In standard clinical practice, even if an orthopedic surgeon suspects a non-union event three months after the occurrence of the fracture, a second surgery is performed no earlier than six months after the primary fracture occurrence. In this context, the possibility of predicting non-union sufficiently early may decrease patient morbidity by means of an earlier medical or surgical intervention. In case of a delayed diagnosis, patients could experience pain for months, may be unable to perform daily living activities, and their professional/working life may be affected as well.

To meet these clinical needs, several studies have investigated the possibility of analyzing circulating bone remodeling biomarkers, whose levels vary depending on bone healing process, and to explore whether BTMs changes are able to predict non-union occurrence. Bone turnover markers include a series of proteins or protein fragments released during bone remodeling [10,11,12,13,14,15,16,17]. BTMs are released into the bloodstream, from which they can be collected and quantified to monitor bone deposition or the resorption rate [10,11,18]. In pathological contexts, the bone-remodeling rate can be altered, leading to changes in circulating factors’ levels. Potential bone turnover markers related to osteoblasts and osteoclasts activities and their role in the remodeling process are described in Figure 1. Additionally, sclerostin (SOST), a glycoprotein mainly produced by osteocytes, and circulating Dickkopf-1 (DKK1) dynamics were recently evaluated in patients after a traumatic long bone fracture. An increase in the postoperative SOST level (without compensatory DKK1 reduction) was found in patients with dysfunctional fracture healing [17]. A systematic review by Breulmann et al. described the role of miRNAs as functional markers in bone remodeling [19]. The authors have shown changes in miR-31-5p, miR-221, and miR-451-5p levels in non-unions, but the heterogeneity of the studies does not yet justify their use in clinical application. 

The aim of this systematic review is to analyze the current knowledge on the association between circulating BTMs levels and the possibility of predicting fracture healing failure in human clinical studies.

## 2. Materials and Methods

This systematic review was performed according to PRISMA guidelines [20] and registered in the “International Prospective Register of Systematic Review” (PROSPERO) in (https://www.crd.york.ac.uk/prospero/display_record.php?ID=CRD42023452744, accessed on 19 August 2023). A literature search was conducted by using PubMed, Cochrane, Cinahl, Web of Science, Scopus and Embase databases. The following combination of terms was used: (“nonunion*” OR “non-union*”) AND (“bone turnover marker*” OR “bone marker*”). No specific restriction on the year of publication was applied. The PICO model (Population, Intervention, Comparison, Outcomes) was used to design this study: (1) studies that considered patients with non-union (Population), (2) with the BTMs evaluation (Interventions), (3) compared/associated with current diagnostic tool of non-union (Comparisons), (4) that reported significant differences (*p* < 0.05) on specific circulating BTMs (Outcomes). Specific inclusion criteria were (1) articles evaluating BTMs in human patients; (2) articles where a control group was considered (healthy subjects or evaluation of the same subject during time). We excluded (1) studies with only in vitro data; (2) studies performed only in animal models; (3) studies where the failure of fracture healing was associated with osteoporosis; (4) studies where the failure of fracture healing was associated with genetic disorders. Additionally, we excluded reviews, conference abstracts, editorials, book chapters, and articles not written in English (full text). All records published until 29 August 2023 were eligible for inclusion.

### Grouping of Studies and Synthesis of Data

The results of the study selection process are summarized in the “Preferred Reporting Items for Systematic Review” (PRISMA) flowchart (Figure 2) showing the consecutive methodological steps of this systematic review. Two of the authors (FP and LR) performed the eligibility assessment independently, in an unblinded, standardized manner. Title and abstract sifts were conducted first (Figure 2), followed by a review of the full text by FP and LR. Only studies fulfilling the eligibility criteria were included. The methodological quality of the selected studies was independently assessed by using standard quality assessment criteria for evaluating research papers [21]. The general characteristics and quality assessment of the included studies and Kmet et al. score were reported in Table 1 [22,23,24,25,26,27,28,29,30,31,32,33,34,35] and S1. As 60% is suggested as a reasonable cut-off, all papers were included in the systematic review. Data were extracted by two reviewers (FP and LR) and tables were created including the information on type and general aim of the study, number of patients, fracture site, age of patients, gender, BTMs analyzed, sample scheduling, and findings (differently expressed BTMs).

## 3. Results

### 3.1. Study Selection

The results of the selection process are summarized in the PRISMA flow chart (Figure 2). Briefly, the initial literature search of six databases retrieved 172 papers and, after the deletion of 89 duplicates, 83 articles were analyzed. A total of 73 documents were excluded, as described in Figure 2. Notably, among the excluded papers, 16 articles were associated with osteoporosis, 4 articles described non-union in patients affected by genetic diseases (hypophosphatasia, osteogenesis imperfecta, and neurofibromatosis), and 3 articles described pseudarthrosis in lumbar/spine fusion.

After the initial selection based on the title and abstract, an additional article was excluded, as it described non-union in patients occupationally exposed to lead. To retrieve any additional studies of interest, from a review of references, five records were added to be considered for eligibility.

### 3.2. Study Characteristics

The study design, aim, and quality assessment of the included studies were described in Table 1 [22,23,24,25,26,27,28,29,30,31,32,33,34,35]. The majority of the studies were based on prospective observational protocols. Two RCTs (randomized controlled trials) were reported, but BTMs evaluation was not the primary objective of these studies; they were focused on the effect of vitamin D or extracorporeal shock wave therapy on bone healing. Two multi-center studies, located in Europe, were reported, while all other studies were monocentric (India *n* = 3, Germany *n* = 3, USA *n* = 2, United Kingdom *n* = 2, Belgium *n* = 1, Italy *n* = 1). The patients’ race/ethnicity and socioeconomic characteristics were not reported in the studies.

### 3.3. Study Reporting

The sample size, site of fracture, age of patients, BTMs assayed, sample scheduling, and main findings were reported in Table 2. Among the studies, a certain heterogeneity in subjects/patients age and site of fractures was observed. Moreover, inconsistencies in sampling schedules make it difficult to compare the BTMs expression levels. To highlight similarities or differences among the studies in terms of a specific BTMs expression over time, findings were reported also by grouping results for each BTM in Table 3. BTMs expression changes were reported by comparing

Healed fracture vs. healthy subjects (different patients and subjects).Healed fracture vs. non-union (different patients).Progression of the fracture healing over time in the same patient (healed or not healed).

The percentage of studies reporting an increased, decreased, or unchanged BTMs level is represented in Appendix A. The results were merged independently from the endpoint of sampling, which can be different among studies.

**Table 2 jcm-13-02333-t002:** Characteristics of the included studies: sample size, site of fracture, gender, age, BTMs analyzed, sampling schedule, number of non-unions, and results.

Reference	Sample Size	Site of Fracture	Gender (%Female)	Age	BTMs	Sampling Schedule	Number of Non-Unions	Findings
[22]	33 NU pz and 35 hc (CTR)	Ulna, radius, humerus, fibula, femur, tibia, clavicle, metatarsus, scaphoid	NU: 31% CTR: 62%	NU: 18–78 (mean: 44) CTR: 23–78 (mean: 32)	Proteomic studies SELDI-TOF-MS and 2D-DIGE	-	33 vs. 35 hc	Up/downregulation in NU vs. CTR: inter-α trypsin inhibitor, hepcidin, S100A8, S100A9, glycated hemoglobin β subunit, PACAP related peptide, complement C3 α-chain, apolipoprotein E, complement C3 and C6 subunits
[23]	102 pz	Femur and tibia; 47% patients have additional fractures in other sites.	31%	18–50 (mean: 23)	CTx, P1NP (serum)	6 and 12 weeks after fracture	20/102	CTx and P1NP (6 weeks) associated with healing at 12 weeks.
[24]	26 DU or NU pz (3 months diagnosis)	Femur, tibia, fibula, humerus, unknown	42%	19–65 (mean: 40)	BAP, CTx, CICP, int-OC, N-Mid OC, OPG, RANKL (serum)	T0 and after 6, 12, and 24 weeks	1/24	BAP level is higher at 6 weeks in patients showing early healing. CICP, int-OC, N-Mid OC levels were lower at 6 weeks in patients that heal after 24 weeks
[25]	36 pz	Tibial plateau fractures	39%	22–73 (mean: 46)	Collagen X (serum)	T0 and after 3, 6, and 12 weeks from treatment	-	Delayed peaks of collagen X expression in patients treated with external fixation or staged open reduction internal fixation
[26]	16 NU pz; 18 and 14 age-matched pz healed within 6 months and 1 month	Long bones	NU: 0% Age-matched pz: 0%	20–39	OPG, RANKL, BAP, OC (serum) and deossipirolidine (DPD) (urine)	-	16 vs. 18–14 matched healed subjects	OPG levels were higher in NU patients compared to CTR No difference in DPD levels in NU vs. healed patients
[27]	49 pz	Radius, humerus.	82%	46–76	Vitamin D3, PTH, BAP, CTx, TRAP5b (serum)	T0, before surgery, and after 1, 4, 8, and 52 weeks	0	No difference in BTM levels over time.
[28]	168 pz	Tibia/fibula	14%	20–79 (mean: 32)	NTX, BSAP (BAP), P1NP and N-Mid OC (serum)	T0 before surgery, and after 8, 12, 24, and 36 e 72 weeks	29/168 DU (6 months) 9/168 NU (12 months)	BAP, P1NP and N-Mid OC levels were lower in DU (at 8–12 and 24 weeks)
[29]	15 NU and 15 healed with similar fractures	Femur, tibia, forearm, humerus	NU: 20% Healed: 20%	20–70 (mean: 46.7 NU) 22–75 (mean: 46.4 healed)	TRACP, P1NP, CTx, BAP (serum)	T0 before surgery, and after 1, 2, 4, 8, 12, 52 weeks	15 NU vs. 15 healed pz	CTx (1 week) is lower in DU, TRACP is lower after 2–4 weeks BAP and P1NP levels: no significant difference in U and NU
[30]	26 DU or NU pz (3 months diagnosis) 13 ONFH pz	Femur, tibia, fibula, humerus, unknown	DU/NU: 42% ONFH: 8%	DU/NU: 19–65 (mean: 40) ONFH: 21–53 (mean: 42)	BAP, CTx, CICP, N-Mid OC, OPG, RANKL (serum)	T0, before surgery, and 12 and 24 weeks, after surgery.	1/24	CICP increase, CTx decrease (good outcome pz)
[31]	20 pz	Tibial shaft fractures treated non-operatively	5%	16–61 (mean: 33.7)	CICP, P1NP, BAP (serum)	1, 4, 8, 14 days and 5, 10, 14, 20 weeks post-fracture	3/20 DU (20 weeks)	CICP lower in DU vs. U at 20 weeks P1NP higher in DU vs. U at 10 weeks BAP DU vs. U no significant difference
[32]	14 pz	tibial shaft fractures	36%	21–70 (mean: 43.7)	CTx, BAP, N-Mid OC (serum)	1, 7, 17, 28, 42, 60, 90, 180, and 365 days post-fracture	4/14 DU	BAP higher in DU vs. U at 1, 26, 52 weeks OC increase is delayed (1 month) in DU vs. U (60th vs. 90th day)
[33]	121 pz and 108 hc	Tibia–fibula fractures	Pz: 12% CTR: 14%	18–45	OC, osteopontin (mRNA and protein in serum)	4, 7, 10, 15, 20, 28 days post fracture	19/121 (DU 24 weeks)	OC higher (protein) higher in U vs. DU at day 20 and 28 Osteopontin no statistical difference among groups
[34]	50 pz	Closed tibial fractures treated non-operatively	22%	16–82 (mean: 30.7)	BAP, OC (in 14 patients) (serum)	0, 2, 4, 6, 8, 10, 12, 14, 16, 18, 20 weeks post fracture	9/50 DU (20 weeks)	OC lower in DU vs. U (8 and 16 weeks) BAP no significant differences
[35]	95 pz	Tibia–fibula fractures treated non-operatively	-	18–45	ALP (serum)	14, 21, 28, 45, 60, and 90 days post fracture	18/95 DU (6 months) 8/95 NU (9 months)	BAP is higher for U > DU > NU (no significant difference)

Legend: BTMs: bone turnover marker; NU: non-union; DU: delayed union; U: union; CTR: controls; pz: patients; hc: healthy controls; ONFH: osteonecrosis of the femoral head.

**Table 3 jcm-13-02333-t003:** Findings related to each BTM in the studies included in the systematic review.

BTMs	Findings	Endpoint of Sampling Showing Significant Difference	Reference
BAP	>in early U	6 weeks	[24]
	↑ in U	12 weeks	[30]
	<in DU	8–12–24 weeks	[28]
	>in DU	1, 26, 52 weeks	[32]
	U vs. DU:NS	5–10–14–20 weeks	[31]
	U > DU > NU but NS	3 weeks	[35]
	U vs. DU:NS	8–16 weeks	[34]
	U vs. NU:NS	-	[26]
	U vs. NU:NS	1–2–4–8–12–52 weeks	[29]
OC	<in DU	8–12–24 weeks	[28]
	<in DU	6 weeks	[24]
	↑ in DU delayed vs. U	60th vs. 90th day	[32]
	>in U	20–28 days (protein by Western blot assay)	[33]
	<in DU	8–16 weeks	[34]
	U vs. NU:NS	-	[26]
CICP	↓ in NU	24 weeks	[30]
	<in DU	6 weeks	[24]
	<in DU	20 weeks	[31]
P1NP	<in DU	6 weeks	[23]
	<in DU	8–12–24 weeks	[28]
	>in DU	10 weeks	[31]
	U vs. NU:NS	1–2–4–8–12–52 weeks	[29]
	<in DU	1 week	[29]
	<in DU	6 weeks	[23]
CTx	↓ in U	12, 24 weeks	[30]
	U vs. NU:NS	6, 12, 24 weeks	[24]
	U vs. NU:NS	1, 4, 8, 52 weeks	[27]
	U vs. NU:NS	1, 2, 4, 7, 12, 25, 52 weeks	[32]
NTX	U vs. NU:NS	-	[28]
TRACP5b	↓ in DU	2–4 weeks	[29]
	U vs. NU:NS	1, 4, 8, 52 weeks	[27]
OPG	>in NU	-	[26]
	U vs. NU:NS Delayed peaks in NU ↑	12–24 weeks 12 (U) vs. 24 (NU) weeks	[24] [30]
RANKL	NS		[30]
	U vs. NU:NS	-	[26]
Collagen X	Delayed peaks with staged open reduction internal fixation	6–12–24 weeks	[25]
Proteomics	< or > for different protein (see text)	-	[22]

Legend: BTMs: bone turnover marker; NU: non-union; DU: delayed union; U: union; NS: no significant difference. < or > refers to changes in respect to control (healed or healthy subject); ↓ or ↑ refers to changes in respect to T0 (surgery) in the same subject/patient.

### 3.4. Analyzed Studies

Among the studies included in this review, the majority of the papers are related to observational studies. Moreover, although Wölfl et al. have performed an RCT, the evaluation of BTMs was a secondary objective of the study and, therefore, sample size evaluation was not calculated on the matter we are interested in, resulting in statistically underpowered results [27]. The reduced sample size of the majority of the evaluated papers has led to only a small number of non-union cases being observed, thus making it difficult to perform a statistically significant evaluation of the prognostic accuracy of the BTMs level.

However, in this systematic review, we were able to identify more than one study analyzing changes in a single BTM level during time and any difference measured between healed and not-healed patients.

Regarding biomarkers able to reflect the formation of new bone, BAP, P1NP, CICP, and OC levels were analyzed. Bone-specific alkaline phosphatase (BAP) is the specific bone isoform of alkaline phosphatase. It is secreted by osteoblasts and it is essential for bone matrix mineralization [36,37]. Granchi et al. explored the expression of BAP in patients with non-unions treated with expanded autologous bone-marrow-derived mesenchymal stromal cells, and these authors observed a higher level of BAP in patients with a good outcome at 6 and 12 weeks [24,30]. According to these data, Kumar et al. found lower BAP levels (8–12–24 weeks) in the serum of patients with delayed union in a study following BTMs expression in 168 patients with tibial fracture treated surgically with intramedullary nailing [28]. However, in this study, the BAP diagnostic accuracy was not satisfying due to the low sensitivity of the BAP level in predicting non-union.

The opposite trend was observed by Herrmann et al., who observed BTMs expression for 52 weeks in 14 patients with tibial shaft fractures who underwent surgery [32]. In this study, the authors found that BAP levels increased significantly in delayed union cases at 1, 25 and 52 weeks. It is worth noting that in this sample size cohort (14 patients), 4 out of 14 patients showed delayed union, but with strong differences. Indeed, two patients developed intramedullary infections, one patient suffered from syringomyelia, and one patient developed uncomplicated non-union.

No significant differences in BAP levels between union and non-union or delayed union cases have been detected by other authors. Indeed, Kurdy evaluated BAP levels in 20 patients with tibial shaft fractures treated non-operatively at 5, 10, 14, and 24 weeks, and no significant difference emerged between delayed union (*n* = 3) and consolidated fracture (*n* = 17) [31]. Fifty patients with closed tibial fractures treated non-operatively were analyzed by Oni et al., and, again, no differences in BAP levels between union and delayed union cases emerged at 8 and 16 weeks [34]. Marchelli et al. and Moghaddam et al. compared BAP levels in patients with non-union with the levels in healed age-matched subjects, and no differences were detected [26,29]. The last study we included in this review focused on alkaline phosphatase, but not in the specific bone isoform. Singh et al. studied 95 patients with tibia–fibula fractures treated non-operatively, a part of which developed delayed union at six months (*n* = 18) and non-union at nine months (*n* = 8) [35]. The levels of alkaline phosphatase were found to be higher in union cases compared to non-union cases at 3 weeks, but the difference was not significant. Thus, so far, conflicting results have been observed for BAP-level changes even when similar sample scheduling was applied.

Osteocalcin is the most abundant non-collagenous protein in the bone matrix, and it is synthetized by osteoblasts [11]. It is embedded in the bone matrix and can be partially released in the bloodstream. The N-terminal/fragment, which is more stable than the intact osteocalcin protein, is usually preferred in serological dosages [38]. In the study by Kumar et al., which involved 168 patients with tibial fracture treated surgically with intramedullary nailing, lower OC levels (8–12–24 weeks) were measured in the serum of patients with delayed union [28]. A similar trend was observed by Granchi et al. in a cohort of 26 patients and found that the OC levels were lower at 6 weeks in patients in whom healing was observed only after 24 weeks (delayed union) [24]. Moreover, Ali et al. compared the expression of OC in 108 healthy control and 121 patients with tibia–fibula fractures, who were followed up to 28 days post-fracture [33]. The expression of OC protein was higher in union vs. delayed union cases on days 20 and 28. According to these data, Oni et al. observed lower levels of OC at 8 and 16 weeks in a small cohort of patients with delayed union [34].

As for BAP measurement, the study by Marchelli et al. did not show any significant difference in OC expression levels in patients with non-union and age-matched subjects with healed fractures [26].

Additionally, no significant differences in OC levels in patients with consolidated fractures or delayed union were observed by Herrmann et al., who followed BTMs expression for 52 weeks in 14 patients with tibial shaft fractures who underwent surgery. The researchers observed that the OC level increase was delayed by 1 month in delayed vs. union cases (on the 60th vs. 90th day after surgery) [32]. Thus, except for the studies in which no significant differences were observed, based both on small cohort of patients, a positive trend between OC levels and bone healing was detected. 

An essential component of the bone matrix is represented by Type I Collagen, whose maturation leads to the release of N and C-term fragments. These propeptides are detectable in patients’ serum, providing a stoichiometric quantitative representation of collagen synthesis [12]. Indeed, decreased levels of CICP at 24 weeks were detected in subjects experiencing non-union, and lower amounts of CICP at 6 weeks was found in patients for whom delayed union was observed at 24 weeks [24,30]. The diagnostic accuracy of CICP was calculated in Granchi et al. and, according to the ROC curve, if the collagen synthesis, as the CICP level, is 10% lower than the baseline, the probability of a poor outcome is very high [30]. However, this result was obtained by analyzing BTM levels in patients treated with a cell therapy approach but who were affected by different diseases (i.e non-union and osteonecrosis of the femoral head). According to Granchi et al., Kurdy found lower levels of CICP in delayed vs. union cases at 20 weeks, in a cohort of twenty patients with tibial shaft fractures treated non-surgically, where three delayed union cases were diagnosed [31].

A similar trend was observed for the N-term fragment of pro-collagen (i.e., P1NP). Levels were found to be lower in delayed union cases at 6 weeks in the study by Stewart et al. (102 patients) and at 8, 12, and 24 weeks in the study by Kumar et al. (168 patients) [23,28]. This study associated a good sensitivity (0.82) and specificity (0.89) for P1NP as a predictor of delayed union [28]. 

On the contrary, in the study by Kurdy, higher levels of P1NP were observed for delayed union cases at 10 weeks in a small cohort of 20 patients with tibial shaft fractures treated non-operatively [31]. In this study, as we described above, an opposite trend for CICP was found. No possible explanation was given for why two fragments derived by procollagen type I were detected with the opposite trend. However, we have yet to determine the stability of these molecules when secreted in circulation, and this may affect their recovery and measurement. 

The last study we analyzed for P1NP expression compared P1NP levels in 15 patients with non-union and 15 healed age-matched subjects, and no significant difference were detected between the two groups in observations conducted for up to 52 weeks [29]. 

Thus, except for the study by Kurdy [31], which reported conflicting data, and the study by Moghaddam et al. [29], for which there was a small sample size, all the other studies found a common trend for collagen synthesis markers, which were found in lower amounts in the blood of patients with delayed or non-existent healing.

In this systematic review, we also considered bone resorption biomarkers, such as N- and C-telopeptides derived from Type I Collagen degradation. During the bone resorption process, CTx and NTX telopeptides are released into circulation, where they can be detected in the bloodstream and in urine [13]. Moghaddam et al. and Stewart et al. associated lower levels of CTx in delayed union cases at 1- and 6-week endpoints, respectively [23,29]. On the contrary, Granchi et al. associated lower levels of CTx with a very high chance of a good outcome at 12 and 24 weeks [30]. The study by Herrmann et al., which focused on 14 patients with tibial shaft fractures, 4 of whom exhibited a delayed union outcome, did not find any difference for CTx in the two groups [32]. Additionally, no differences in CTx levels over time were observed by Wölfl et al., but, in this study, as X-rays were available only for 50% patients, the sample size was reduced [27]. Additionally, regarding markers related to collagen degradation, only the study by Kumar et al. evaluated NTX expression, and no significant differences emerged for the serum levels of NTX at any endpoint analyzed [28].

An additional marker related to bone resorption is the enzyme TRACP5b, which is produced solely by osteoclasts and is involved in their resorption activity [39]. Significantly decreased TRACP5b levels were reported by Moghaddam et al. in non-union patients two and four weeks post-surgery [29]. As for CTx markers, the study by Wölfl et al. did not find any difference in the TRACP5b level between the union and delayed-union group [27]. However, this study is under-powered as, after 1 year, X-rays were available for only 50% of the studied patients.

The interaction between osteoblast and osteoclast cells is a crucial aspect of the bone remodeling process, and the RANKL/OPG/RANK pathway is involved in regulating the differentiation of precursors into multinucleated osteoclasts. Indeed, the RANKL/OPG ratio in bone marrow is an important determinant of bone mass in normal and disease states [40].

The changes in circulating levels of RANKL and OPG were evaluated in the study of Marchelli et al. by comparing patients with non-union in long bones with age-matched subjects who healed within 1 or 6 months, while in their study, Granchi et al. evaluated RANKL changes in non-union and delayed union patients treated with expanded mesenchymal stromal cells (MSC), as a regenerative approach. Neither study revealed any significant change in RANKL levels between groups or over time [26,30]. Meanwhile, Marchelli et al. observed a significantly higher amount of OPG in the non-union patients group compared to subjects in whom normal union was achieved after 1 or 6 months [26]. 

Granchi et al. have shown that the increase in OPG levels after treatment is significant for patients with good outcomes at 12 weeks, and is delayed at 24 weeks for patients who experience treatment failure after surgery [30]. However, no significant changes between groups were detected at any endpoints [24,30]. It would have been interesting to evaluate the RANKL/OPG ratio in addition to single-factor changes, as their activity is strictly dependent on such changes. Therefore, we suggest considering this aspect in future studies design.

Bone remodeling also involves other proteins and molecules. Working et al. considered Collagen X as a potential circulating biomarker for bone fracture healing [25]. In this study, 36 patients with tibial plateau fractures were treated non-operatively, or with immediate open-reduction internal fixation or with staged open-reduction internal fixation. The authors noted a delay in the Collagen X peak in patients with staged fixation, and have suggested an association between this biomarker level and delayed loading. No differences were associated with patients’ outcomes.

In this systematic review, we also reported the findings of de Seny et al., who analyzed circulating markers expression with completely different techniques based on proteomics [22]. This approach enabled the identification of new potential biomarkers that were down- or upregulated in the serum of healthy volunteers compared to atrophic non-union patients. Indeed, the authors detected differences in the expression of inter-α-trypsin inhibitor H4, hepcidin, S100A8, S100A9, glycated hemoglobin β subunit, PACAP related peptide, complement C3 α-chain, and apolipoprotein E in the serum of both non-union and healthy control subjects. However, the comparison between non-union and healthy subjects presented some limitations. Indeed, some differences revealed in this study, such as hepcidin expression, may be associated with the different inflammatory status typical of non-union patients.

## 4. Discussion 

Non-union diagnosis is currently based on radiographic and clinical evaluation, but the amount of literature concerning the relationship between bone turnover markers changes in the bloodstream and the fracture-healing process is increasing. The aim of this current systematic review was, therefore, to systematically screen the literature in order to verify if a consensus has been reached regarding the usefulness of BTMs evaluation to predict or support non-union diagnosis and/or bone healing.

Overall, in this systematic review, significant heterogeneity was detected among studies investigating the association between BTMs levels and fracture healing and/or non-union. Heterogeneity was observed in terms of the patient population (e.g., the age of patients, gender, fracture sites, and the type and severity of the fracture), the type of treatment (e.g., surgical, non-surgical, or a regenerative approach, such as MSC treatment), the sampling (e.g., sample scheduling or fasting testing), and the methodology used to measure BTMs levels (e.g., ELISA assay, proteomics, or Western blot assay). The significant heterogeneity of the included studies did not allow us to perform a meta-analysis.

General inclusion and exclusion criteria were quite homogeneous among the studies included in this systematic review. Indeed, the majority of the studies considered comorbidities known to interfere with the bone-repair process (i.e., diabetes mellitus, anemia, malnutrition, peripheral vascular disease, and hypothyroidism) and the use of some drugs (i.e., corticosteroids, non-steroidal anti-inflammatory drugs, antibiotics) among the exclusion criteria. Moreover, patients were excluded from the studies in the presence of infection and/or open fracture, as they are considered, per se, to be relevant factors associated with delayed union risk.

However, patients may have more than one fracture at different sites. In these studies, although the outcome was based only on the main fracture, the level of circulating BTMs could be influenced by bone healing that also occurs at the other fracture sites. Thus, this may represent a confounding aspect in the analysis of BTMs levels’ association with bone healing. 

In this systematic review, we excluded studies related to bone fractures in patients affected by osteoporosis. A reduced bone density characterizes osteoporotic bone, which has a higher incidence in older people and in post-menopausal women [41]. Osteoporosis can be considered a systemic skeletal disease, which is characterized by reduced bone mass and qualitative alterations of bone macro and micro-architecture. In patients with this condition, changes in circulating BTMs levels may derive from an impaired bone turnover that is not limited to fracture site. Moreover, biphosphonates have a direct effect on bone resorption markers [42,43]. Thus, we considered osteoporosis as a peculiar disease that has to be analyzed separately.

The studies included in this systematic review showed some limitations. Indeed, healthy controls were used as a comparator group, and in most studies, the sample size was small, with few non-unions, resulting in statistically under-powered results. Thus, obtaining significant differences useful for calculating the predictive value of BTMs was quite difficult.

Careful evaluation was also necessary when results were obtained with cohorts that differed in terms of gender. Indeed, even at the basal level, differences in circulating BTMs levels have been reported among healthy subjects (e.g., pre-menopausal females, post-menopausal females, and males) [30,44,45]. Moreover age, regional differences, ethnicity, exercise, diet, and the menstrual cycle may be associated with sources of variability in BTMs levels [46,47].

According to this variability, the use of a specific reference range for BTMs measurement requires careful evaluation and an in-depth knowledge of the sources of variability and of the population analyzed. Thus, it is worth underlining that the BTM level at a given time-point is multi-factorial and may have a high level of variation in the general population. For these reasons, measuring changes inside the same patient can be more meaningful.

This systematic review has highlighted that further research is required to clarify and validate the presented findings on BTMs before they can be suggested as clinical markers of bone healing and/or as predictors of the non-union occurrence. Moreover, in the experimental design of new studies, it will be worth considering some important factors related to BTMs expression. Indeed, a marked circadian variation has been shown in some BTMs. Changes in CTx and OC expression with nighttime or early morning peaks have been highlighted [18,48,49]. On the contrary, BAP, P1NP, OPG, RANKL, and sclerostin did not appear to be significantly influenced by circadian rhythms [10,48]. Moreover, fasting has been shown to reduce CTx circadian variations, while it does not affect P1NP and OC expression [48]. Due to this recognized variability, the National Bone Health Alliance has suggested some tips to improve pre-analytical standardization of samples with the aim of improving the reliability and interpretation of data [50]. Pre-analytical variability comprises controllable and uncontrollable factors. Among controllable determinants, to prevent the interference of circadian variations, the authors suggested collecting the samples in the early hours of the morning. Moreover, for BTMs levels affected by food intake, they suggested collection should take place after an overnight fast. Among uncontrollable determinants, the choice of the appropriate reference group must consider the patients’ age, gender, and menopause status. Additionally, incorrect storage of samples may significantly alter the BTMs levels [51]. It is relevant to consider all these aspects during sampling and to report this information in the study methodology.

Moreover, by considering the ideal characteristics of clinically relevant non-union biomarkers, we suggest focusing future investigation on BTMs that showed significant variation at early time points, such as during the peri-operative period or up to 12 weeks after the operation, at which time radiographic measures are not sufficiently predictive.

## 5. Conclusions

Despite the heterogeneity of the studies reported in the literature, which was associated with patients’ characteristics (fracture site, gender, age) and with the sample schedule or type of analysis performed to monitor BTMs, this systematic review highlighted potential bone remodeling markers (BAP, CICP, P1NP, CTx, OC, TRACP5b, and OPG), whose levels appeared to change according to the fracture healing process. At present, there are not enough data to suggest or promote the use of a single BTM in association with radiological and clinical evaluation of patients for earlier detection of non-union. However, the improvement of the knowledge on the biological variability of BTMs and future clinical studies with higher samples number and the inclusion of detailed patient and sampling characteristics will help us to better define BTMs changes during the bone healing process.

## Figures and Tables

**Figure 1 jcm-13-02333-f001:**
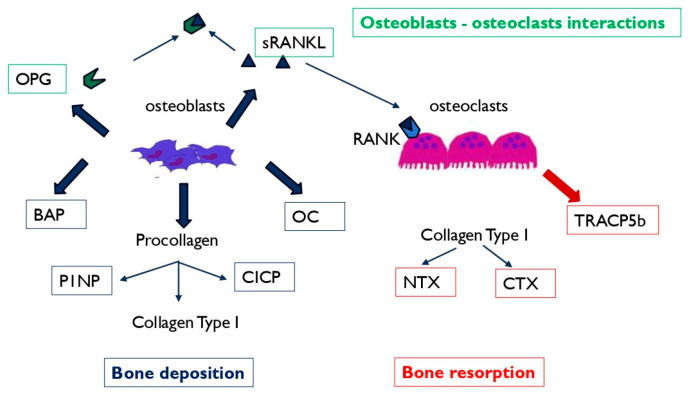
Schematic representation of the biological source of circulating bone turnover markers (BTMs). During the bone deposition process, the osteoblasts secrete bone alkaline phosphatase (BAP) and osteocalcin (OC). N-terminal (P1NP) and C-terminal propeptide (CICP) are released from procollagen-Type 1. Type-I-Collagen degradation leads to C-terminal telopeptide (CTx) and N-terminal telopeptide (NTX) release. Osteoclasts participate in bone matrix degradation by means of tartrate-resistant acid phosphatase 5b enzyme (TRACP5b). The receptor activator of nuclear factor-κB ligand (RANKL) can be bound by the decoy receptor osteoprotegerin (OPG), which prevents RANKL from binding to its receptor activator of nuclear factor-κB (RANK), inhibiting osteoclastogenesis.

**Figure 2 jcm-13-02333-f002:**
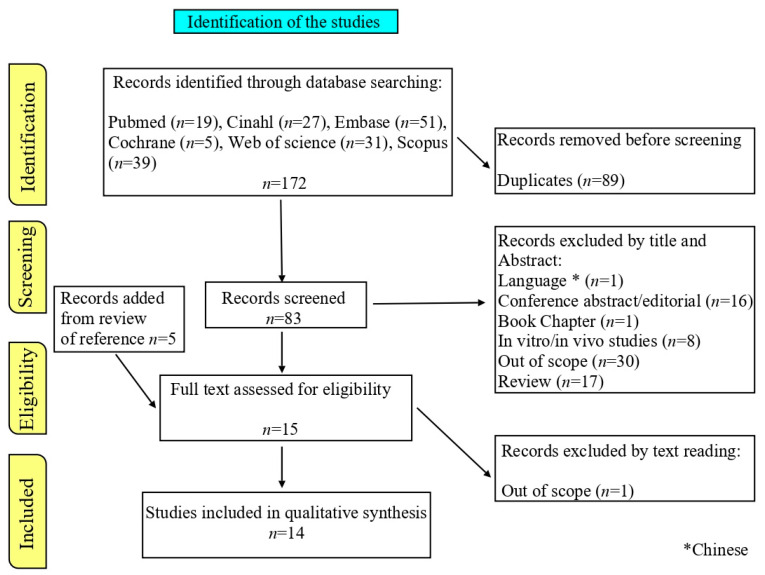
Identification and selection of studies: PRISMA flow chart.

**Table 1 jcm-13-02333-t001:** Characteristics and quality assessment of the included studies.

Reference	Country of Publication	Aim of the Study	Design of the Study	Qualitative Score *
[22]	Belgium	Comparison between BTM levels’ (non-union vs. healthy subjects)	Observational	84%
[23]	USA	Evaluation of BTM levels’ changes over time	RCT (patients treated with or without vitamin D)	89%
[24]	Multi-center (Italy, Germany, France, Spain)	Evaluation of BTM levels’ changes over time	Prospective, controlled	89%
[25]	USA	Evaluation of Collagen X levels’ changes over time	Prospective longitudinal study	74%
[26]	Italy	Comparison between BTM levels’ (fractured vs. healed subjects)	Observational study	80%
[27]	Germany	Evaluation of BTM levels’ changes over time	RCT and prospective longitudinal study for BTMs evaluation	69%
[28]	India	Evaluation of BTM levels’ changes over time	Prospective observational study—Level of evidence 2	83%
[29]	Germany	Evaluation of BTM levels’ changes over time	Prospective open control study	93%
[30]	Multi-center (Italy, Germany, France, Spain)	Evaluation of BTM levels’ changes over time	Prospective, controlled, phase 2 trial (treatment with culture-expanded MSC)	92.5%
[31]	United Kingdom	Evaluation of BTM levels’ changes over time	Prospective observational study	92.5%
[32]	Germany	Evaluation of BTM levels’ changes over time	Prospective observational study	92.5%
[33]	India	Evaluation of BTM levels’ changes over time	Prospective observational study	93%
[34]	United Kingdom	Evaluation of BTM levels’ changes over time	Prospective observational study	62%
[35]	India	Evaluation of BTM levels’ changes over time	Prospective observational study	60%

* Kmet et al. [21].

## Data Availability

Data sharing is not applicable to this article, as no datasets were generated during the current study.

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
