# Peer review of "Association between Bone Turnover Markers and Fracture Healing in Long Bone Non-Union: A Systematic Review"

_jcm, 2024, doi:10.3390/jcm13082333_

Round 1

Reviewer 1 Report

Comments and Suggestions for Authors

Thank you for this useful systematic review on bone turnover markers in fracture healing.  This review could be enhanced by some minor amendments as follows:

Introduction:

Line 50: amend smoke habit to current smoking.  Also consider adding diabetes for impaired fracture healing and reference.

Page 2 section 1.2  Please add that bone turnover markers are linked to circadian rhythm in some cases and that they should be taken fasting and first thing in the morning to blunt this effect.

Methods

Line 182 add A in front the literature...

Please also include a rationale for excluding osteoporosis studies - was this due to bisphosphonates?

Line 204 

Please add the agreement between the reviewers for scoring the papers and how many were involved in each stage of the process including title / abstract screen / full text screen.

Results

Line 356 please amend whose to which

Line 384 revealed should be reveal

Line 406 - are healthy controls an appropriate comparator group?  Please bring this through to the discussion.  

Discussion

The discussion is a little short and could be enhanced by expanding the comments on study design, power of the studies (some are small and probably under-powered and also the baseline heterogeneity of BTMs in the general population.

Please add consideration that BTMs have a high level of variation and are generally useful for measuring changes, but for one off measurements are much less useful.  

line 448 - circadian rhythm links have been known for a long time - please remove the word recent from this sentence.

Comments on the Quality of English Language

Generally good with a few corrections as above

Author Response

Reviewer 1:

Thank you for this useful systematic review on bone turnover markers in fracture healing.  This review could be enhanced by some minor amendments as follows:

Introduction:

Line 50: amend smoke habit to current smoking. Also, consider adding diabetes for impaired fracture healing and reference.

According to the reviewer, we modified the text and we added the following reference for diabetes and fracture non-union risk.

Negus OJ, Watts D, Loveday DT. Br J Hosp Med (Lond). Diabetes: a major risk factor in trauma and orthopaedic surgery. 2021, 82(1):1-5. doi: 10.12968/hmed.2020.0609.

Page 2 section 1.2  Please add that bone turnover markers are linked to circadian rhythm in some cases and that they should be taken fasting and first thing in the morning to blunt this effect.

According to the reviewer, we modified the text in section 1.2.

Methods

Line 182 add A in front the literature...

According to the reviewer we modified the text.

Please also include a rationale for excluding osteoporosis studies - was this due to bisphosphonates?

We decided to exclude patients with osteoporosis because bisphosphonates have a direct effect on bone resorption markers (Eastell R et al, 2017; Ashcherkin N et al, 2023; ). To clarify this point, we added the reference.

Eastell, R.; Szulc, P. Use of bone turnover markers in postmenopausal osteoporosis. Lancet Diabetes Endocrinol. 2017, 5, 908-923. doi: 10.1016/S2213-8587(17)30184-5

Ashcherkin N, Patel AA, Algeciras-Schimnich A, Doshi KB. Bone turnover markers to monitor oral bisphosphonate therapy. Cleve Clin J Med. 2023, 90(1):26-31. doi: 10.3949/ccjm.90a.22002.

Line 204

Please add the agreement between the reviewers for scoring the papers and how many were involved in each stage of the process including title / abstract screen / full text screen.

We added in supplementary data a new table (Table S1) describing the scoring of paper assigned by each reviewer and some more details in the methods section (paragraph 2.1). In details, two of the authors (FP and LR) performed the eligibility assessment independently, in an unblinded, standardized manner. Title and abstract sift were conducted first (see Figure 2), followed by review of full text by FP and LR. Only studies fulfilling the eligibility criteria were included.

Results

Line 356 please amend whose to which

According to the reviewer, we modified the text.

Line 384 revealed should be reveal

According to the reviewer, we modified the text.

Line 406 - are healthy controls an appropriate comparator group? Please bring this through to the discussion.  

According to the reviewer, we modified the text and we added comments on “3.4 Analyzed studies” and “discussion” paragraphs.

Discussion

The discussion is a little short and could be enhanced by expanding the comments on study design, power of the studies (some are small and probably under-powered and also the baseline heterogeneity of BTMs in the general population.

According to the reviewer, we expand the discussion with additional comments.

Please add consideration that BTMs have a high level of variation and are generally useful for measuring changes, but for one off measurements are much less useful.

According to the reviewer we add this comment.

Line 448 - circadian rhythm links have been known for a long time - please remove the word recent from this sentence.

According to the reviewer, we modified the text.

Comments on the Quality of English Language

Generally good with a few corrections as above

Reviewer 2 Report

Comments and Suggestions for Authors

It's honor of me to review your interesting manuscript. I have some suggestions.

1. Please ensure the inclusion of the most recent studies to provide the latest insights into bone turnover markers and fracture healing. Highlight any new biomarkers or emerging technologies being explored in this context.

2. Please provide more specific details about the search strategy, including search terms, databases, and the date range of the literature search. Clarify the process for study selection and data extraction, including how discrepancies were resolved.

3. Please include a summary table that presents the quality assessment of each included study, possibly using a standardized tool or checklist. This will help readers understand the strength of the evidence presented.

4. Please enhance the clarity and impact of the results section with additional visual aids, such as charts, graphs, and tables. These could illustrate the key findings, differences in BTM levels, and patterns related to fracture healing or non-union.

5. Please Elaborate on the potential clinical applications of your findings, discussing how BTMs could be integrated into existing diagnostic or prognostic frameworks for fracture healing. Suggest practical guidelines or considerations for clinicians.

Comments on the Quality of English Language

The quality of English language appears to be good, with coherent sentence structure and appropriate scientific terminology. However, a meticulous proofreading for typographical errors, grammar, and clarity could further refine the manuscript, ensuring that it is communicated effectively to an international audience.

Author Response

It's honor of me to review your interesting manuscript. I have some suggestions.

  1. Please ensure the inclusion of the most recent studies to provide the latest insights into bone turnover markers and fracture healing. Highlight any new biomarkers or emerging technologies being explored in this context.

According to reviewer, we added recently data related to circulating Dickkopf-1(DKK1) and sclerostin (SOST) dynamics after human traumatic long bone fracture. Starlinger J et al (2024) reported an increased postoperative SOST level (without compensatory DKK1 reduction) in patients (below 50 years) with dysfunctional fracture healing. Moreover, we described the role of some miRNA as functional markers in bone remodeling, but the heterogeneity of the studies does not yet justify their use in clinical application (see “1.2.4. Other BTMs markers” paragraph).

According to PRISMA guidelines, in this systematic review, we specify the date of last search in the consulted databases (August 29th 2023).

Starlinger J, Santol J, Kaiser G, Sarahrudi K. Close negative correlation of local and circulating Dickkopf-1 and Sclerostin levels during human fracture healing. Sci Rep. 2024 Mar 19;14(1):6524.doi: 10.1038/s41598-024-55756-5.

  1. Please provide more specific details about the search strategy, including search terms, databases, and the date range of the literature search. Clarify the process for study selection and data extraction, including how discrepancies were resolved.

The search strategy has been described in ‘Materials and Methods’ section. The literature search was conducted in PubMed, Cochrane, Cinahl, Web of Science, Scopus and Embase databases. The following combination of terms was used: ("nonunion*" OR "non-union*") AND ("bone turnover marker*" OR "bone marker*"). All records published until August 29 th 2023 were eligible for inclusion.

Study selection process is summarized in the “Preferred Reporting Items for Systematic Review” (PRISMA) flowchart (Figure 2). We added additional information in “2.1. Grouping of studies and synthesis of data” paragraph. In Table S1 we described the quality assessment evaluation procedure. We did not observe relevant discrepancies in the reviewers evaluation. Any disagreement between reviewers was resolved by consensus.

  1. Please include a summary table that presents the quality assessment of each included study, possibly using a standardized tool or checklist. This will help readers understand the strength of the evidence presented.

We added in supplementary data a new table (Table S1) describing the scoring of paper given by each reviewer and some, more details in the methods section (paragraph 2.1).

  1. Please enhance the clarity and impact of the results section with additional visual aids, such as charts, graphs, and tables. These could illustrate the key findings, differences in BTM levels, and patterns related to fracture healing or non-union.

According to reviewer, we represented the proportion of studies reporting an increased, decreased or unchanged BTMs level as a pie chart in Fig S1.

  1. Please Elaborate on the potential clinical applications of your findings, discussing how BTMs could be integrated into existing diagnostic or prognostic frameworks for fracture healing. Suggest practical guidelines or considerations for clinicians.

The translation into clinical practice on how BTMs could be integrated into existing diagnostic or prognostic frameworks for fracture healing requires a standardized approach starting from reliable data. In this systematic review, the heterogeneity of reported data, sampling and patients characteristics highlighted that results on BTMs changes to improve early detection of failure or signs of slow bone regeneration need further validation. Moreover, individual factors with influence on BTMs levels should be thoroughly investigated and reported. Moreover, by considering the ideal characteristics of a clinical relevant non-union biomarkers, we suggest to focus future investigation on BTMs that showed significant variation at early time points, such as peri-operative period or up to 12 weeks, when radiographic measure are not sufficiently predictive.

Reviewer 3 Report

Comments and Suggestions for Authors

The manuscript by Perut et al. is a systematic review examining the relationships between bone turnover markers and fracture healing in the context on non-unions. This is an interesting topic that would provide valuable insight. However, the manuscript as it stands is very difficult to read and unorganized. The manuscript will require extensive revision before it is suitable for publication, with a special focus on English editing. Specific concerns are listed below:

1. The introduction is very lengthy and provides a very detailed overview of BTMs. This is more appropriate for the discussion. 

2. The PROSPERO link states that the study has been registered, but it states that the study has not be started. This should be updated accordingly. 

3. For exclusion criteria, it should be made clear that in vivo studies are referring to animal studies as human clinical studies are in vivo. 

4. It is unclear how many individuals performed the search. 

5. Table 2 is extremely difficult to interpret. I recommend the authors separate out the sample sizes, fracture characteristics, number of nonunions in unique columns. Then simplify the changes in the BTMs in a separate column. It would also be useful to include the sex of individuals in the study.

6. The discussion is very difficult to follow and only repeats the results. The discussion requires extensive revision.

Comments on the Quality of English Language

The grammar and English require extensive editing. 

Author Response

The manuscript by Perut et al. is a systematic review examining the relationships between bone turnover markers and fracture healing in the context on non-unions. This is an interesting topic that would provide valuable insight. However, the manuscript as it stands is very difficult to read and unorganized. The manuscript will require extensive revision before it is suitable for publication, with a special focus on English editing. Specific concerns are listed below:

  1. The introduction is very lengthy and provides a very detailed overview of BTMs. This is more appropriate for the discussion. 

According to the reviewer suggestions, we modified the introduction.

  1. The PROSPERO link states that the study has been registered, but it states that the study has not be started. This should be updated accordingly. 

We agree with the reviewer and we have updated the PROSPERO link.

  1. For exclusion criteria, it should be made clear that in vivo studies are referring to animal studies as human clinical studies are in vivo. 

We modified the text in paragraph 2 “materials and methods” according to the reviewer suggestions to clarify exclusion criteria.

  1. It is unclear how many individuals performed the search. 

We added in supplementary data a new table (Table S1) describing the scoring of paper given by each reviewer and some, more details in the methods section (paragraph 2.1).

  1. Table 2 is extremely difficult to interpret. I recommend the authors separate out the sample sizes, fracture characteristics, number of nonunions in unique columns. Then simplify the changes in the BTMs in a separate column. It would also be useful to include the sex of individuals in the study.

According to the reviewer suggestions, we modified Table 2, We separated columns (sample sizes, fracture characteristics, number of nonunions) and added the gender of individuals involved in the studies.

  1. The discussion is very difficult to follow and only repeats the results. The discussion requires extensive revision.

According to the reviewer suggestions, we modified the discussion section.

Comments on the Quality of English Language

The grammar and English require extensive editing.

English has been revised.

Round 2

Reviewer 3 Report

Comments and Suggestions for Authors

The authors did not address my concerns related to the introduction which is still too long. In fact, they only deleted a couple of sentences. The introduction should serve as a brief introduction to the study, not a detailed discussion on every BTM and bone remodeling. This should be included in the discussion to frame the results within the biology. It is also unclear what the authors mean by "BTM dosage" (Line 504). The authors also mention "some tips to improve pre-analytical standardization" however no tips are every discussed. 

Comments on the Quality of English Language

There are grammatical errors remaining. Moreover, the overuse of "etc..." should be avoided in the discussion. 
